# Sn catalyst reconstruction and microenvironment modulation for efficient amino acid electrosynthesis via C−N coupling

Shuhe Han [1,8], Huimin Liu[2,3,8], Janis Timoshenko [2], Joonbaek Jang[2], Mengyao Su[1], Chenghua Sun [4], Chengying Guo[5], Yanmei Huang[5], Arno Bergmann [2], Beatriz Roldan Cuenya [2], Yifu Yu [5], Bin Zhang [5,6], Kai Leng [1] & Kian Ping Loh [1,7] ✉

The electrosynthesis of amino acids represents a fascinating and promising frontier in green chemistry, offering a sustainable alternative to conventional industrial processes such as the energy-intensive Strecker synthesis through the adoption of efficient, electricity-driven methods. Herein, Sn is identified as an effective catalyst for glycine electrosynthesis using concentrated nitric acid and oxalic acid as feedstocks, and we investigate the reaction mechanism at industrial-level current rate (1 A cm⁻²). In-situ characterization reveals that the Sn undergoes dynamic valence cycle and reconstructs into amorphous-Sn under acidic conditions. At high current, the change in local pH promotes the anionic states of oxalic acid and C-intermediates, which enhances the adsorption of key intermediates such as glyoxalic acid and acid oxime. This switches the mechanism from a chain reaction to an interfacial hydrogenation, thereby increasing the rate of glycine formation. By increasing the dominance of interfacial reaction versus the chain reaction, we achieve a glycine Faradaic efficiency of 93%, and industrial-level partial current density of 0.9 A cm⁻² in a flow cell.

Amino acids, as the fundamental building blocks of life, play vital roles in pharmaceutical production and function as key nutritional additives in food[1]. Commercial synthesis methods, such as the Strecker reaction and enzymatic or microbial processes, face significant limitations[2,3]. While enzymatic or microbial approaches offer specificity, their reliance on complex protocols and lack of versatility restricts broader application[4]. The Strecker reaction, although widely adopted, depends on toxic cyanide reagents and ammonia derived from the energy-intensive Haber−Bosch process, increasing sustainability and safety concerns[5]. These challenges underscore the urgent

need for innovative, environmentally benign synthesis strategies powered by clean energy to align with global sustainability goals.

In recent years, the electrocatalytic coreduction of NOx and COx/ small organic molecules to achieve C−N coupling has emerged as a promising green synthesis strategy[6–9]. For the electrosynthesis of amino acids, operating in concentrated acids offers significant advantages. These include facilitating the formation of crucial hydroxylamine intermediates and enabling the direct use of HNO₃ as both a reactant and electrolyte, which streamlines downstream separation and reduces costs. Furthermore, acidic conditions are compatible with

---

[1]Department of Applied Physics, The Hong Kong Polytechnic University, Hong Kong, China. [2]Department of Interface Science, Fritz-Haber Institute of the Max-Planck Society, Berlin, Germany. [3]College of Electronic Information and Optical Engineering, Nankai University, Tianjin, China. [4]Department of Chemistry and Biotechnology, Swinburne University of Technology, Hawthorn, VIC, Australia. [5]Department of Chemistry, School of Science, Institute of Molecular Plus, Tianjin University, Tianjin, China. [6]International Joint Laboratory of Low-carbon Chemical Engineering of Ministry of Education, Tianjin University, Tianjin, China. [7]Department of Chemistry, National University of Singapore, Singapore, Singapore. [8]These authors contributed equally: Shuhe Han, Huimin Liu. ✉e-mail: chmlohkp@nus.edu.sg

plasma-catalyzed $N_2$ oxidation to generate the nitrogen reactant, paving the way for a more economical and sustainable synthesis pathway[10]. However, this approach faces considerable challenges. The acidic environment suffers from strong competition from the hydrogen evolution reaction (HER) and restricts catalyst selection due to the corrosive nature of concentrated nitric acid. These limitations currently make it difficult to simultaneously achieve both high Faradaic efficiency (FE) and high yield rates. Consequently, strategies for direct $HNO_3$ conversion to construct C−N bonds remain elusive. Importantly, the lack of in situ characterization evidence renders the reaction mechanisms unclear−particularly in terms of identifying the active species, understanding the reactant states within the catalyst surface microenvironment, and elucidating their transformation pathways. As a result, research often relies on trial-and-error methodologies, limiting the rational design of efficient catalysts. Thus, an in-depth understanding of the electrocatalytic active sites and reaction mechanism in complex processes is crucial[11,12].

Herein, we achieved the electrosynthesis of glycine (Gly) by the co-reduction of $HNO_3$ and oxalic acid (OA). $HNO_3$ can be sourced from $NO_x$ gas wet scrubbing liquid from industrial exhaust, whereas OA can be derived from $CO_2$ reduction or biomass conversion[13,14]. By choosing Sn as the catalyst, we achieved an FE of 65.9% and a partial current density of 659 mA cm$^{-2}$ for electrosynthetic Gly at −1.4 V vs. SHE (corresponding to −1.37 V vs. RHE) in a H-cell, showing its competitiveness over most metal catalysts. The results of kinetic studies revealed the existence of two reaction mechanisms: a chain reaction (low FE of Gly) and a direct interfacial hydrogenation mechanism (high FE of Gly), with the latter dominating the reaction following multiple cycles of catalyst reconstruction. In situ X-ray Diffraction (XRD), in situ Raman, in situ X-ray absorption spectroscopy (XAS), differential electrochemical mass spectrometry (DEMS) show that Sn and $HNO_3$ can undergo a spontaneous redox reaction, converting to $SnO_x$ and NO, respectively. Simultaneously, SnOx can be electroreduced to Sn metal, forming a dynamic valence cycle. The dynamic reconstruction of Sn and SnO causes amorphization. Interestingly, the reaction efficiency improved after this amorphization period; thus, the active species catalyst should be amorphous-Sn. At industrial-level current densities, the rotating ring disk electrode test shows local H$^+$ concentration differs from that in the bulk solution by two orders of magnitude. This means that interfacial weak acid reactant molecules (OA and key C-intermediates) exist as anionic species. Electrochemical quartz crystal microbalance and density functional calculation results show that these factors contribute to enhanced catalyst-reactant/intermediate adsorption energies, causing the mechanism to switch from a chain reaction to direct interfacial hydrogenation. By controlling the local pH and using amorphous Sn as a catalyst, the reaction can be optimized such that it is controlled by interfacial kinetics; this increases the FE to 93% and the yield rate to 2.18 g h$^{-1}$ for the one-step synthesis of Gly (corresponding to an industrial-level partial current density of 0.9 A cm$^{-2}$).

## Results

### Electrocatalyst screening and preparation
Based on existing literature[15–17], the electrosynthesis of glycine (Gly) can be divided into four key steps (Fig. 1a and Supplementary Note 1). We screened common transition metal catalysts and operated industrial-level current densities (0.1 A cm$^{-2}$) under acidic conditions, which are optimal for Gly synthesis (Fig. 1b). Among the screened catalysts, Sn foil exhibits the highest performance (R1 – R4, Fig. 1c and Supplementary Table 1), owing no less to the formation of an insoluble oxide layer that passivates it (Supplementary Note 2) in concentrated $HNO_3$. Other transition metals either do not form an insoluble oxide layer or do not form an oxide layer at all; thus, they are corroded in $HNO_3$. Here, we employed a modified electro-deposition method to fabricate Sn nanoparticles (NPs) on carbon

paper (CP) under ambient conditions. Structural characterization by scanning electron microscopy (SEM), high-resolution transmission electron microscopy (HRTEM), and XRD confirmed the particulate morphology of crystalline Sn (Fig. 1d−f and Supplementary Fig. 1). Sn K-edge X-ray absorption near-edge structure (XANES) and $k^2$-weighted Fourier transform extended X-ray absorption fine structure (FT − EXAFS) data further corroborated the metallic nature of Sn (Fig. 1g, h).

### Electrocatalytic performance test in H−cells
The specific routes for the Gly electrosynthesis are outlined in Fig. 2a and involve three critical intermediates: $NH_2OH$, glyoxylic acid (GOA), and glyoxylic acid oxime (GAO)[7]. The electrochemical performance was evaluated in a three-electrode system, with product quantification by $^1H$ nuclear magnetic resonance (NMR, Supplementary Figs. 2, 3 and Supplementary Note 3). Rotating disk electrode (RDE) measurements revealed that the introduction of $HNO_3$ and/or OA significantly increased the reduction current density ($j$, Supplementary Fig. 4), indicating accelerated proton and electron transfer kinetics for the N- and/or C-reaction reduction reactions (N-/C-/(N + C)-RR) compared with HER. Notably, the coreduction system outperformed the individual N-RR or C-RR processes in terms of $j$ values.

The potential-dependent FE of all products is summarized in Supplementary Fig. 5. For C−N coupling product, at the optimal potential −1.4 V vs. SHE (corresponding to −1.37 V vs. RHE), its FE is 72% (GAO+Gly, Fig. 2b). Among them, the Gly FE is 66%, and its yield rate is 2.07 mmol cm$^{-2}$ h$^{-1}$ (Fig. 2c), corresponding to a partial $j$ of 667 mA cm$^{-2}$ (Supplementary Fig. 6), highlighting competitive performance (Supplementary Table 2)[8,18]. Furthermore, the system demonstrated robust performance across a wide concentration range of $HNO_3$ and OA (0.1−1 M) and maintained stability for 70 h (Fig. 2d, e), showing its potential for scalable application.

### Evaluation and analysis of substrate scope
We investigated the generality of our co-reduction strategy by substituting various N- and C- sources (Table 1). Replacing $NO_3^-$ with NO or $NO_2^-$ also yielded Gly, demonstrating that $NO_x$ waste gases or liquids are viable N sources for this approach (Supplementary Fig. 7 and Supplementary Note 4). However, when oxalic acid (OA) was replaced with other monoprotic or nonconjugated carboxylic acids, such as formic acid, acetic acid, benzoic acid, or succinic acid, no corresponding C − N coupling products formed (Supplementary Fig. 8). Control experiments revealed that their carboxyl (−COOH) groups could not be reduced to aldehydes (Supplementary Fig. 9). Further experiments with aldehydes or ketones (e.g., formaldehyde, acetaldehyde, and acetone) as carbon source yielded oximes rather than amines (Supplementary Fig. 10). These results suggest that the electroreduction of −COOH to −C = O is an essential step in the construction of the C − N bond process, and the electron-withdrawing effect of the neighboring −COOH group in OA likely governs the hydrogenation behavior of the $\alpha$ − COOH and $\alpha$ − C = N− bond, ruling out direct nucleophilic addition mechanism proposed by other reports (such as nucleophilic attack by $NH_2OH$ or $NH_x$ species on the −COOH to form −C − N− bond). In contrast, water-soluble $\alpha$-keto acids, including glyoxylic acid (GOA), pyruvate, and $\alpha$-ketoglutarate, were successfully converted to their respective amino acids (Gly, alanine, glutamate). Notably, $\alpha$-ketoglutarate achieved an 85% FE for the glutamate synthesis, a critical neurotransmitter in the vertebrate nervous system, highlighting the utility of this strategy for electrosynthesized amino acids (Supplementary Fig. 11). Essentially, their functional group transformation pathways are similar, and the steps are simpler compared to the $HNO_3$ + OA system. These results suggest that our electrosynthetic approach is broadly applicable to the production of water-soluble amino acids, leveraging electrochemical activation of $\alpha$-keto acid precursors.

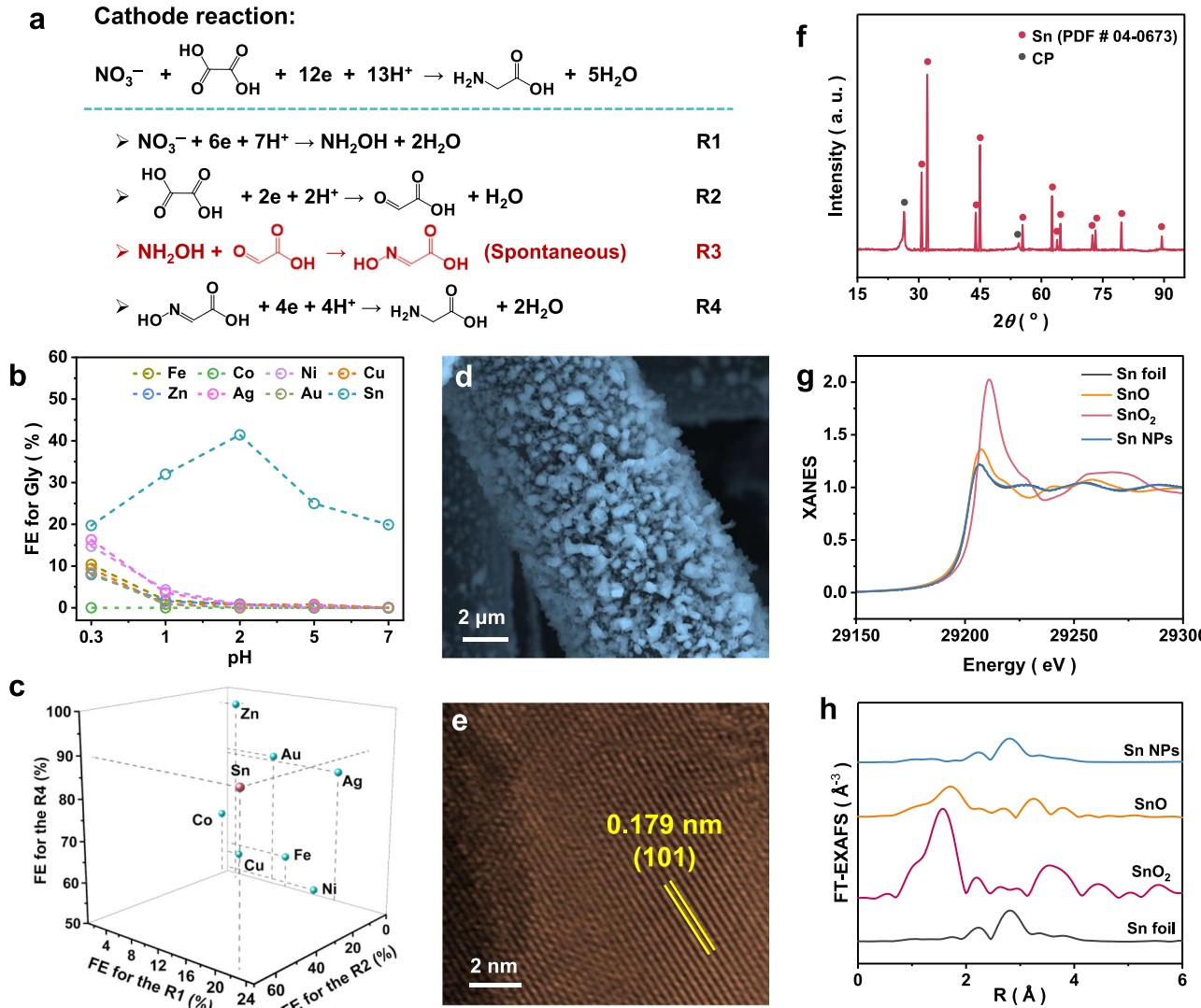

**Fig. 1 | Screening and synthesis of electrocatalysts. a** Sub-reactions involved in Gly electrosynthesis, including R1: electrochemical nitrate reduction to hydroxylamine; R2: electrochemical reduction of oxalic acid to glyoxalic acid (GOA); R3: spontaneous coupling of hydroxylamine with GOA to form glyoxylic acid oxime (GAO); R4: electrochemical reduction of GAO to Gly. **b** FE for Gly electrosynthesis from HNO₃ and OA using common transition metal catalysts in solutions with varying pH. **c** FE of common transition metal foils (purity > 99%) used as catalysts for different sub-reactions at a current density of 100 mA cm⁻². **d** SEM image of self-supported Sn NPs on CP. **e** HRTEM image of Sn nanoparticles. **f** XRD pattern of self-supported Sn NPs on CP. **g, h** Sn K-edge XANES and FT−EXAFS spectra of Sn NPs compared with reference materials. All electrocatalytic potentials reported were not corrected for *iR* drop. Source data for Fig. 1 are provided as a Source Data file.

## Mechanism investigation

**Reaction pathway analysis.** Time-dependent $^1$H NMR analysis (Fig. 3a–c) reveals distinct trends for Gly and three key intermediates (NH₂OH, GOA, and GAO). Initially, the intermediates and Gly appear sequentially (Fig. 3d), and the Gly FE slowly increases from zero (Fig. 3e), which is consistent with a classic chain reaction mechanism (Supplementary Note 5). However, between 0.5 and 1 h, the concentration of the intermediates GOA and GAO sharply decreases (Fig. 3f and Supplementary Fig. 12), and the GOA concentrations become undetectable within -0.5 to 1 h. In contrast, Gly production accelerates, and its FE and concentration increases. This phenomenon cannot be explained by the chain mechanism. Since the concentration of the reactant changes slightly during this process (from 0.5 M to >0.4 M), we can rule out the effect of decreased reactant concentrations on the rate of intermediate generation. The divergence above suggests a shift in the reaction mechanism from a chain-based propagation in the aqueous phase to a direct interfacial hydrogenation—a pathway that may still go through GOA/GAO intermediates but without their desorption from the catalyst surface or accumulation into the electrolyte (Supplementary Note 6). Therefore, the electrocatalytic performance observed in the H-cell is attributed to the synergistic interplay between these two mechanisms.

Moreover, we also investigated the small amount of oxamic acid (OMA) produced in the mixed system, because some researchers consider it a key intermediate for the Gly generation. The electrocatalytic results showed that OMA was hardly electroreduced to Gly (Supplementary Fig. 13). Thus, it may be a byproduct of the Beckmann rearrangement process from GAO rather than an effective intermediate for Gly synthesis.

**Identification of catalytically active species.** The transition from a chain mechanism to direct interfacial hydrogenation occurs between 0.5 and 1 h. We therefore investigated the corresponding changes in catalyst state within this critical time window. After electrosynthesis, SEM and ICP−OES analyses revealed that the Sn neither detached nor dissolved (Supplementary Fig. 14). However, STEM and XRD indicated amorphous transformation of the Sn

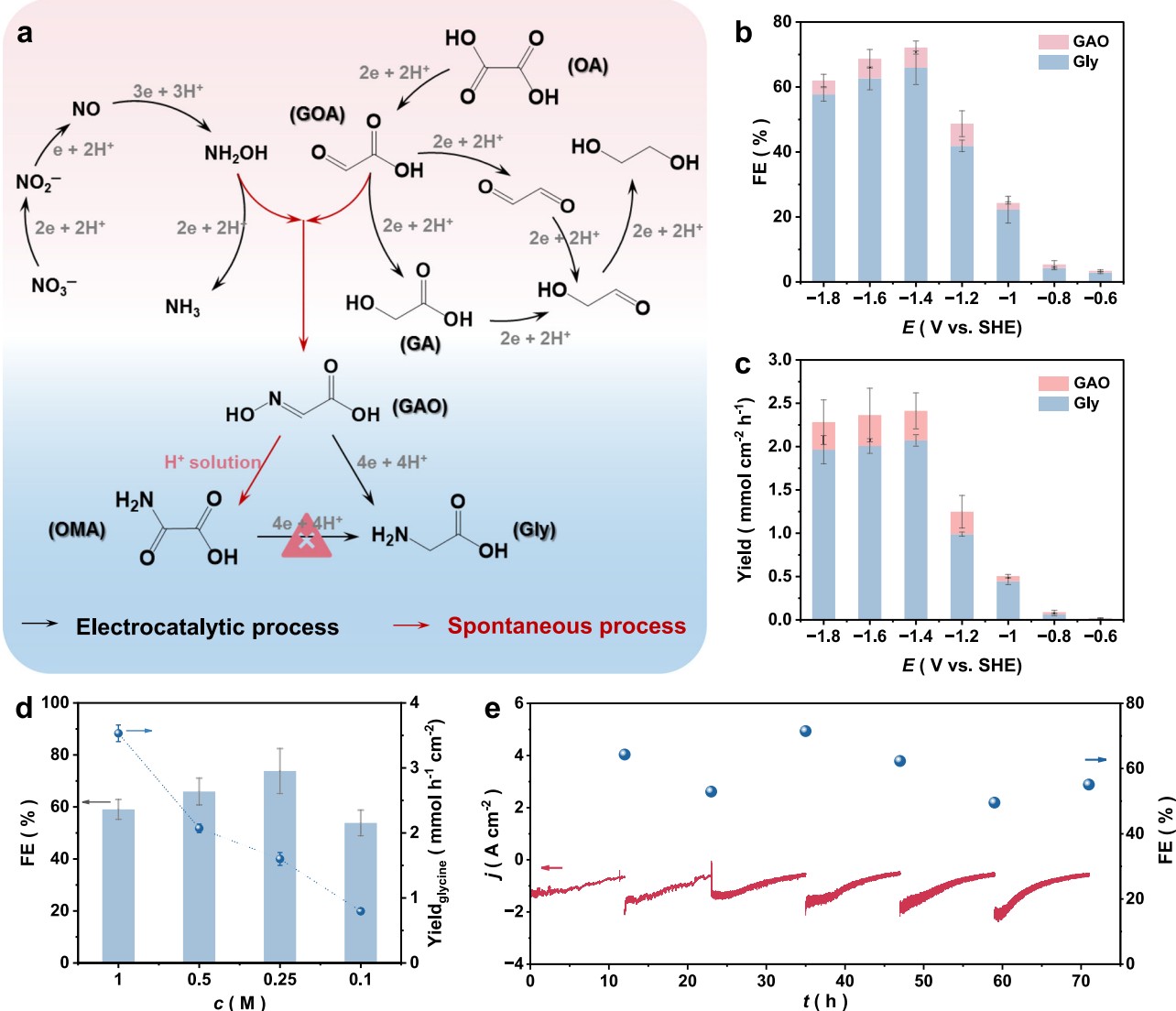

**Fig. 2 | Electrocatalytic performance for Gly synthesis. a** Schematic illustration of Gly synthesis using $HNO_3$ and OA as reactants. **b** FE of C – N coupling products, including Gly and GAO. **c** The yield rates of C – N coupling products, including Gly and GAO. The electrolyte is 0.5 M $HNO_3$ + 0.5 M OA, and the volume is 8 mL. **d** FE and yield rate of Gly over Sn at different concentrations of $HNO_3$ and OA, and the electrolyte volume is 8 mL. **e** Long-term electrocatalytic stability test of synthetic Gly at −1.4 V vs. SHE. The electrolyte volume was 40 mL. The disconnected current curve is caused by the replacement of the electrolyte. Source data for Fig. 2 are provided as a Source Data file.

catalyst (Supplementary Fig. 15). In situ XRD showed that Sn amorphization was complete within the first 30 min (Fig. 4a). According to the Pourbaix diagram, Sn should be in the metallic state at −1.4 V vs. SHE[19]. In situ Raman and CV curve also supported this inference (Fig. 4b and Supplementary Fig. 16). However, XPS results exhibited an increase in $Sn^{II}$ after electrochemistry testing (Supplementary Fig. 17). Intermittent in situ Raman (Fig. 4c and Supplementary Figs. 18–20) and DEMS (Fig. 4d and Supplementary Figs. 21, 22) revealed that, without an applied potential, Sn can undergo a spontaneous reaction with $HNO_3$ to produce $SnO_x$ and NO (Supplementary Note 7)[20–22]; and this characteristic peaks of $SnO_x$ disappear at −1.4 V vs. SHE. This suggested a dynamic Sn valence cycling, arising from the interaction between the catalyst and reactants, and involving both a non-electrochemical redox and electrochemical reduction process. This behavior, analogous to oxidation–redeposition, gradually induce lattice destabilization, driving the crystalline–to–amorphous transition. In-situ XAS also supports this conclusion. In nitrate-containing electrolyte, the proportion of $SnO_x$ (SnO and $SnO_2$) is observed to be high, even at an applied potential of −1.4 V vs. SHE (Fig. 4d, e;

Supplementary Figs. 23, 24 and Supplementary Note 8). Given the complexity of the catalyst composition, the actual active sites are likely $Sn^0$, $SnO_x$, or $Sn/SnO_x$ interfaces. We compared the Bode plots and electrocatalytic performance of amorphous Sn and standard samples (Sn, SnO, and $SnO_2$). The Bode plots and LSV curves show that among the three Sn-species ($Sn^0$, $Sn^{2+}$, and $Sn^{4+}$), amorphous $Sn^0$ is the real active site (Fig. 4g; Supplementary Fig. 25 and Supplementary Note 9)[23]. From a thermodynamic perspective, elements with standard electrode potentials similar to or negative to that of Sn may undergo a similar catalytic valence cycle mechanism during the electrosynthesis of glycine in an acidic environment, highlighting the importance of monitoring catalyst dynamic reconstruction to reveal the true active sites.

Furthermore, we investigated the effect of catalyst amorphization on the chain reaction and interfacial reaction mechanism. In situ electrochemical quartz crystal microbalance (EQCM) showed that amorphous Sn binds GOA and GAO intermediates more readily (Fig. 4h; Supplementary Fig. 26 and Supplementary Note 10), which enhances the rate of the subsequent direct hydrogenation process.

## Table 1 | Substrate scope for different N- and C-sources

| Substrate | FE of C–N coupling products (%) | | Sum (C–N coupling prodcuts) |
|---|---|---|---|
| | R'–C(=N–OH)–R | R'–CH(NH$_2$)–R | |
| KNO$_3$ [a] | 6.2 | 65.9 | 72.1 |
| KNO$_2$ [a] | 13.6 | 52.8 | 66.4 |
| NO [a] | 16.2 | 67.9 | 84.1 |
| R–COOH [b] (R = –H/–CH$_3$/–Ph/ –CH$_2$CH$_2$COOH) | / | / | / |
| R–C(=O)–R' (R and R' = –H/–CH$_2$) | 76.6% (Formaldehyde oxime) 79.5% (Acetaldehyde oxime) 81.3% (Acetone oxime) | / | Same as their oxime |
| glyoxylic acid (O=CH–COOH) | 23.8 | 59.5 | 83.3 |
| pyruvic acid (O=C(CH$_3$)–COOH) | 26.9 | 62.6 | 89.5 |
| 2-oxoglutaric acid (HOOC–CH$_2$–CH$_2$–C(=O)–COOH) | 4.8 | 85.3 | 90.1 |

OA was used as a C source with different N sources. HNO$_3$ was used as the N-source with different C sources

[a] In this reaction system, a mixture of 0.25 M H$_2$SO$_4$ + 0.5 M OA was used as the electrolyte to keep the pH of the system the same as that in other experiments. KNO$_3$, KNO$_2$ or NO (99.99%) was used as the N source.

[b] Only NH$_2$OH and organic reactants were detected in these experimental results.

Bader charge analysis revealed a lower surface charge on amorphous Sn compared to crystalline Sn (−0.27e vs. crystalline Sn: −0.57e, Fig. 4i), which reduces electrostatic repulsion with O-containing molecules. Moreover, amorphous Sn has high disorder, showing defects, cavity and lowly co-ordinated Sn (Supplementary Data 1–4). These features not only generate surface charge fluctuation but also provide abundant active sites (acting as Lewis acid sites), which facilitate the formation of stronger Sn–O bonds between Sn and O-containing organic molecules. While on crystalline Sn (101), the rigid lattice and uniform charge distribution restrict geometric adaptation, leading to suboptimal bonding. Thus, Sn amorphization facilitates the direct hydrogenation mechanism.

**Reactants and microenvironment evolution.** In situ ATR−IR spectroscopy was employed to track the evolution of C- and N- species during the reaction (Fig. 5a). The spectral peak assignments were supported by standard references (Supplementary Fig. 27) and single-component systems (N-RR and C-RR; Supplementary Fig. 28).

At reductive potential, the characteristic peaks of the intermediates (such as NO, NO$_2$, NH$_2$OH, GOA, GAO, and Gly, etc.) increase, proving that these intermediates are generated according to the reaction pathway illustrated in Fig. 2a[8,18,24–26]. Considering that both the catalyst reconstruction and the switching of reaction mechanism

occurred after 30 min, we performed a time-dependent ATR−IR test at the optimal potential (Fig. 5b), and the IR difference spectrum can clearly show the changes of characteristic peaks over time. The two obvious ranges of peaks (1100–1270 cm$^{-1}$ and 1620–1680 cm$^{-1}$) represent the same species as in Fig. 5a, suggesting that intermediates are shared between the chain and interfacial hydrogenation mechanisms. In contrast, new peaks at 1700–1750 cm$^{-1}$ gradually appear after ~30 min (Fig. 5b and Supplementary Fig. 29), corresponding to the different molecular forms of OA (OA: 1745 cm$^{-1}$, OA$^-$: 1725 cm$^{-1}$, OA$^{2-}$: 1720 cm$^{-1}$)[27], which implies a large change in the local pH.

We carried out rotating ring disk electrode test to monitor the local pH of the catalyst surface (Supplementary Fig. 30). By using the reduction current of the Pt ring electrode as a probe, the local pH can be quantified using the standard curve method (Supplementary Fig. 31, 32 and Supplementary Note 11). The results show that at the optimal potential, the local pH (~2.5) over the amorphous Sn is approximately three times that of the bulk solution (Fig. 5c, d and Supplementary Note 12). Since the reactants OA and the intermediates (GOA and GAO) are weak acids (the ionization pH is about ~1), according to the Henderson-Hasselbalch equation, the molecules that actually participate in the reaction should exist as anions (Supplementary Note 13). The electrochemical impedance spectroscopy (EIS) results revealed that the kinetic limitation of the co-reduction process may be related to the C-species generated (Supplementary Fig. 33). In-situ EQCM results showed that intermediate anion species are more readily adsorbed on amorphous Sn surfaces (Fig. 5e and Supplementary Fig. 34), especially for GAO. The DFT results showed that anionic species showed the more negative adsorption energy compared with that of the neutral molecule on amorphous Sn (Fig. 5f and Supplementary Data 5–12). Due to the limitations of modeling charged adsorbates using DFT, the calculated dissociation energies are used to indicate qualitative trends rather than to determine absolute reaction energetics. However, the adsorption energy on the crystalline Sn surface shows an opposite trend (Supplementary Fig. 35). These results suggest that local pH alkalization is another key factor in controlling the reaction mechanism.

**Scaled-up of Gly synthesis in flow reactors and techno-economic analysis (TEA).** Based on the above findings, the Gly synthesis reaction is suggested to proceed through two mechanisms. The interfacial pathway has a higher FE and reaction rate than the chain reactions. The transition from a chain to interfacial hydrogenation mechanism arises from catalyst amorphization and elevated local pH. Although this shift occurs spontaneously, accelerating it should enhance performance. Operating in conventional H-cell systems, even with prolonged reaction times or electrolyte replacement, failed to optimize the interfacial pathways. To address this, a flow cell system (10 cm$^2$ electrode area; Fig. 6a and Supplementary Fig. 36) employing pre-activated amorphous Sn was designed. By regulating flow rate and reactant ratios to control local pH, interfacial hydrogenation dominated rapidly. At 10 A (1 A cm$^{-2}$) and 30 mL min$^{-1}$ flow rate, the system achieved a 93% FE and 2.18 g h$^{-1}$ Gly yield (0.9 A cm$^{-2}$ partial current), surpassing H-cell performance (Supplementary Fig. 37). The high FE obtained after the mechanistic transition (Fig. 6b) contrasts sharply with the results obtained in H-cell systems (Fig. 3), validating the strategy. However, the increased glycolic acid (GA) byproduct formation (Supplementary Fig. 38) highlighted mass transfer limitations in scaled-up systems. While using a higher flow rate could mitigate this, practical constraints prompted alternative optimization: adjusting the HNO$_3$/OA ratio and extending reaction time to 15 min. This trade-off reduced the FE to 72.16% but achieved a 98% OA conversion and 91% selectivity (C-basis; Supplementary Fig. 39). Since our system does not contain any additional electrolyte salts, after a simple rotary evaporation, 0.29 g of Gly (Fig. 6c) with a purity of 97.6% can be obtained. According to a basic techno-economic analysis[10], the energy consumption for

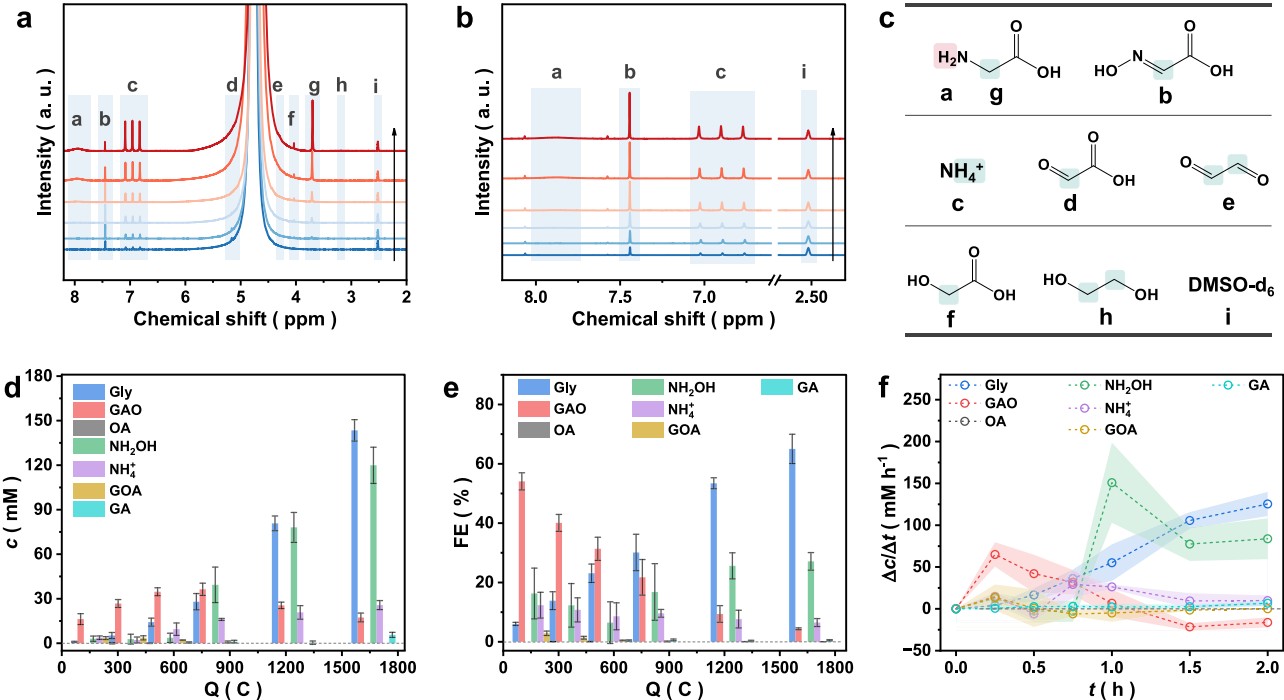

**Fig. 3 | Time-dependent product monitoring over Sn. a** $^1$H NMR spectra of the electrolytes after different reaction times. The initial electrolyte was 0.5 M HNO$_3$ and 0.5 M OA. **b** $^1$H NMR spectra of the electrolytes with 0.2 mL of GOA added after different reaction times. The purpose was to capture NH$_2$OH and calculate the NH$_2$OH concentration with GAO concentrations between (**b**) and (**a**). The quantification of NH$_4^+$ was based on (**b**) to prevent interference from NH$_2$OH

decomposition. **c** Attribution of the characteristic NMR peaks of (**a**) and (**b**). **d**, **e** FEs and concentrations of various products according to the (**a**) and (**b**) results. **f** The average concentration change rate at different reaction times, which is the ratio of the concentration difference to the time difference between two consecutive test results. **f** is extracted from the data in (**d**). Source data are provided as a Source Data file.

producing Gly is 0.2498 MW ton$^{-1}$. The cost of obtaining Gly by our system (1002.13 USD ton$^{-1}$) is half that of the commercial price (2060.44 USD ton$^{-1}$). Coupling the electrochemical flow system with low-temperature plasma technology to generate N-reactant may improve the yield further[10]. The product of plasma catalysis (using air and water as reactants) is HNO$_3$ and HNO$_2$, which can be directly introduced as feedstock in the electrosynthesis system without any addition of salts, highlighting the potential scalability of this approach.

In summary, we have demonstrated Gly electrosynthesis under ambient conditions using amorphous Sn as catalysts and HNO$_3$ and OA as reactants. The co-reduction process involves a transition from a less efficient chain reaction to a more efficient interfacial hydrogenation reaction for Gly electrosynthesis, accompanied by the Sn catalyst amorphization process and an increase in pH in the microenvironment. The dynamic valence cycling of Sn catalyst reconstructs it into an amorphous state that is more catalytically active than the crystalline state. An increase in local pH alters the ionization state of organic molecules (including the reactant OA and key intermediates GOA and GAO). Both of these factors change the adsorption/desorption energy of key intermediates, thereby switching the mechanism from a chain reaction to an interfacial hydrogenation reaction. Scale-up experiments guided by the above findings achieved an optimal FE of 93% and an industrial-level partial $j$ of -0.9 A cm$^{-2}$ for the synthesis of Gly. This work not only provides an efficient method for Gly synthesis but highlights the importance of exploiting catalysts reconstruction and changes in catalytic pathways during electrosynthesis.

## Methods
### Materials
All the chemicals were purchased from Sigma-Aldrich and were of analytical grade and were used as received without further purification. Deionized water (DIW) was used in all the experiments.

### Material characterization
Scanning electron microscopy (SEM) images were taken with a field emission scanning electron microscope (Tescan MIRA). Transmission electron microscopy (TEM) images were obtained via a Spectra 300 (S) TEM. X−ray diffraction (XRD) patterns were recorded with a Rigaku SmartLab 9 kW diffraction system by using a Cu $K\alpha$ radiation source ($\lambda$ = 0.154178 nm). X−ray photoelectron spectroscopy (XPS) data were collected on a Thermo Fisher Scientific Nexsa instrument. All the spectra were collected at a vacuum pressure of <2 × 10$^{-7}$ Pa, and the cumulative scanning number of each sample was 20. The sample was prepared and transferred into an Ar environment to avoid oxidation. All binding energies were referenced to the C 1s peak at 284.6 eV[28]. An inductively coupled plasma optical emission spectrometer (ICP–OES) was used with an Agilent 7700x instrument. The ultraviolet–visible (UV–Vis) absorbance spectra were measured on a PERKIN ELMER_UV–Vis–NIR spectrophotometer. Fourier transform infrared (FTIR) spectroscopy was carried out with a BRUKER_Fourier transform infrared spectrometer. The Raman spectra were obtained on a Renishaw Micro−Raman Spectroscopy System with 532 nm laser light with a power of 20 mW. Nuclear magnetic resonance (NMR) spectra were recorded on a Jeol ECZ500R 500 MHz solid−state NMR spectrometer ($^1$H NMR). The pH values of the electrolytes were determined via a pH meter (pH−100B, LICHEN, China).

### Preparation of self-supported Sn nanoparticles
The self-supported Sn loaded on carbon paper was prepared via modified electrodeposition methods. Using a three-electrode system, carbon paper was used as the working electrode, Ag/AgCl was used as the reference electrode, and a Sn plate was used as the counter electrode. At −1 V vs. Ag/AgCl potential, self-supported Sn can be obtained after 20 min of electrodeposition in 0.5 M H$_2$SO$_4$ + 0.1 M SnCl$_2$ mixed solutions. After electrodeposition, the carbon paper was rinsed with DIW to remove unstable Sn nanoparticles.

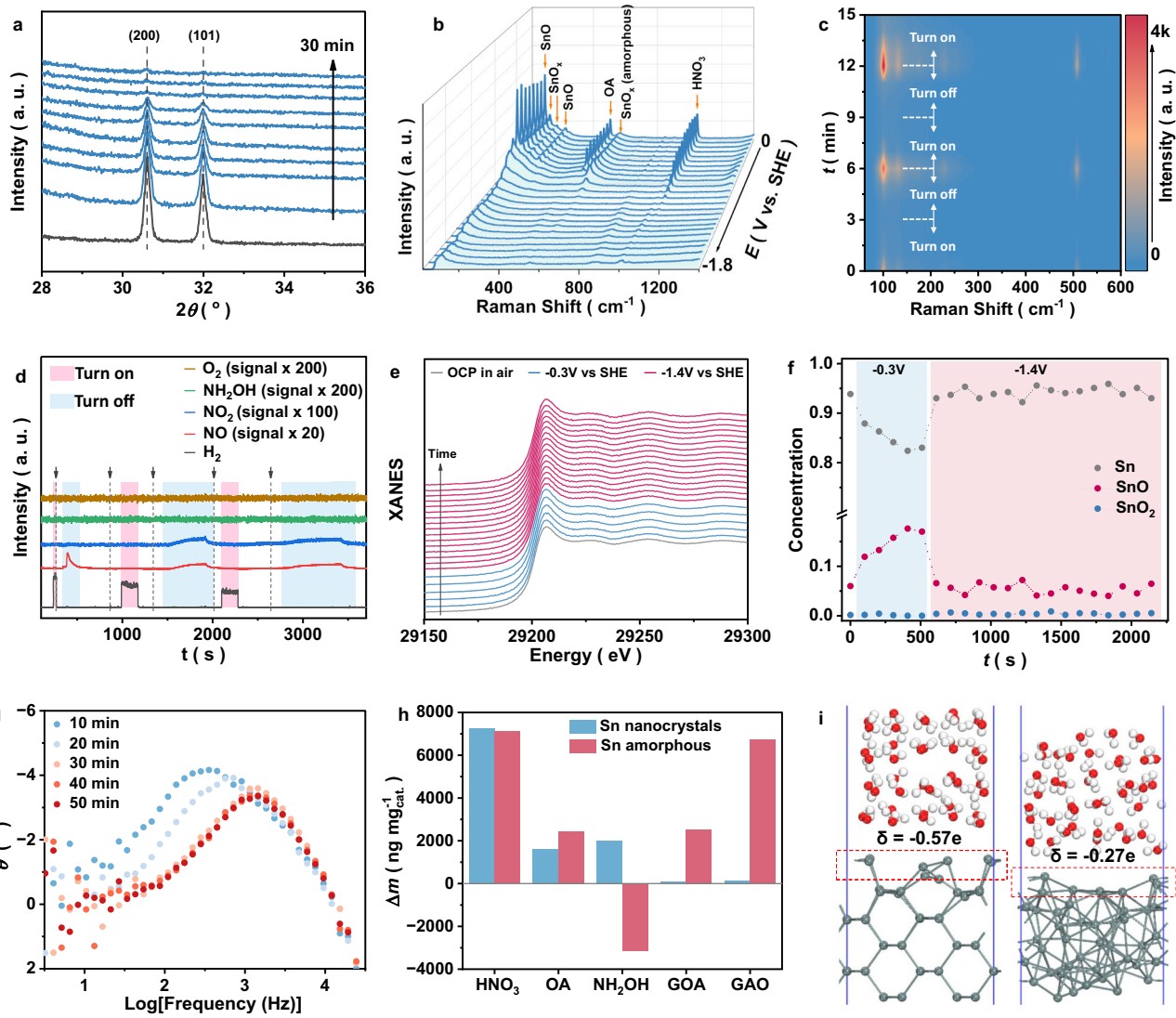

**Fig. 4 | Exploration of active sites. a** In situ XRD patterns of the Sn catalysts during the electrosynthesis of Gly. The black curve was obtained without an applied potential, and the blue curves were obtained at −1.4 V vs. SHE. and **b** In situ Raman tests of the Sn catalysts during the electrosynthesis of Gly. The experimental potential ranges from 0 V to −1.8 V vs. SHE. **c** Intermittent in situ Raman tests with alternating conditions at a turn-on (−1.4 V vs. SHE) and a turn-off voltage. **d** Intermittent in situ DEMS measurement in alternating 0.5 M HNO₃ and 0.25 M H₂SO₄ electrolytes. The initial electrolyte tested was 0.25 M H₂SO₄. The dashed lines and arrows indicate the switching of electrolytes. The red area represents 0.25 M H₂SO₄ at −1.4 V vs. RHE. The blue area represents 0.5 M HNO₃ without any potential. **e** In situ XANES data at different potentials. The gray line represents the test curve obtained in air. The blue lines represent the test curve obtained in the mixed electrolyte (0.5 M HNO₃ + 0.5 M OA) at −0.3 V vs. SHE. The red lines represent the test curve obtained in the mixed electrolyte

(0.5 M HNO₃ + 0.5 M OA) at −1.4 V vs. SHE. **f** In situ valence analysis of Sn based on linear combination analysis of in situ XANES data. The blue and red areas correspond to the blue and red curves in Fig. 4d, respectively. **g** Bode plots for Sn nanoparticles with different amorphous ratios. The amorphous ratio is regulated by the different reaction times of the crystalline Sn nanoparticles in the mixed electrolyte. **h** In situ EQCM test for mass changes (Δm) in crystalline and amorphous Sn at different potentials. Figure 4h shows a summary of the maximum Δm. The electrolyte was a 0.5 M solution of the corresponding tested chemicals. The potential range is −0.4 V to −1.6 V vs. SHE. Amorphous Sn was obtained after electroactivation for 30 min in the mixed electrolyte over crystalline Sn at −1.4 V vs. SHE. **i** Calculated Bader charges δ on top layers (range of the red dotted box) for amorphous and crystallized Sn (101), with O, Sn and H are shown with red, dark gray and white balls. Source data for Fig. 4 are provided as a Source Data file.

## Electrochemical measurements

Electrochemical measurements were performed with an CHI−760E workstation (Ivium Technologies B.V.). A typical three−electrode H −cell was used, which included a working electrode (Sn nanoparticles grown on carbon paper), a Ag/AgCl electrode (saturated KCl solution) as the reference electrode, and a carbon rod counter electrode in the electrolyte (the H⁺ concentration is approximately 0.5 M); these electrodes were separated into a cathode cell (8 mL) and an anode cell (8 mL) by a proton exchange membrane. The initial electrolyte pH was 0.45 ± 0.02 (Error bars indicate the pH fluctuation values of different batches of prepared solutions). For catalytic potential, we did not use

iR correction, except when special instructions were used. For the chronoamperometry test, carbon paper (0.5 × 0.5 cm²) decorated with Sn nanoparticles was used as the working electrode. The current density was normalized by the geometric area of the electrode. All the electrochemical data (except for the stability test data) were repeated more than 3 times, and the error bars represent the standard deviation of the data. All potentials were calibrated to the standard hydrogen electrode (SHE) by the following equation:

$$E_{SHE} = E_{Ag/AgCl} + \varphi_{reference}$$

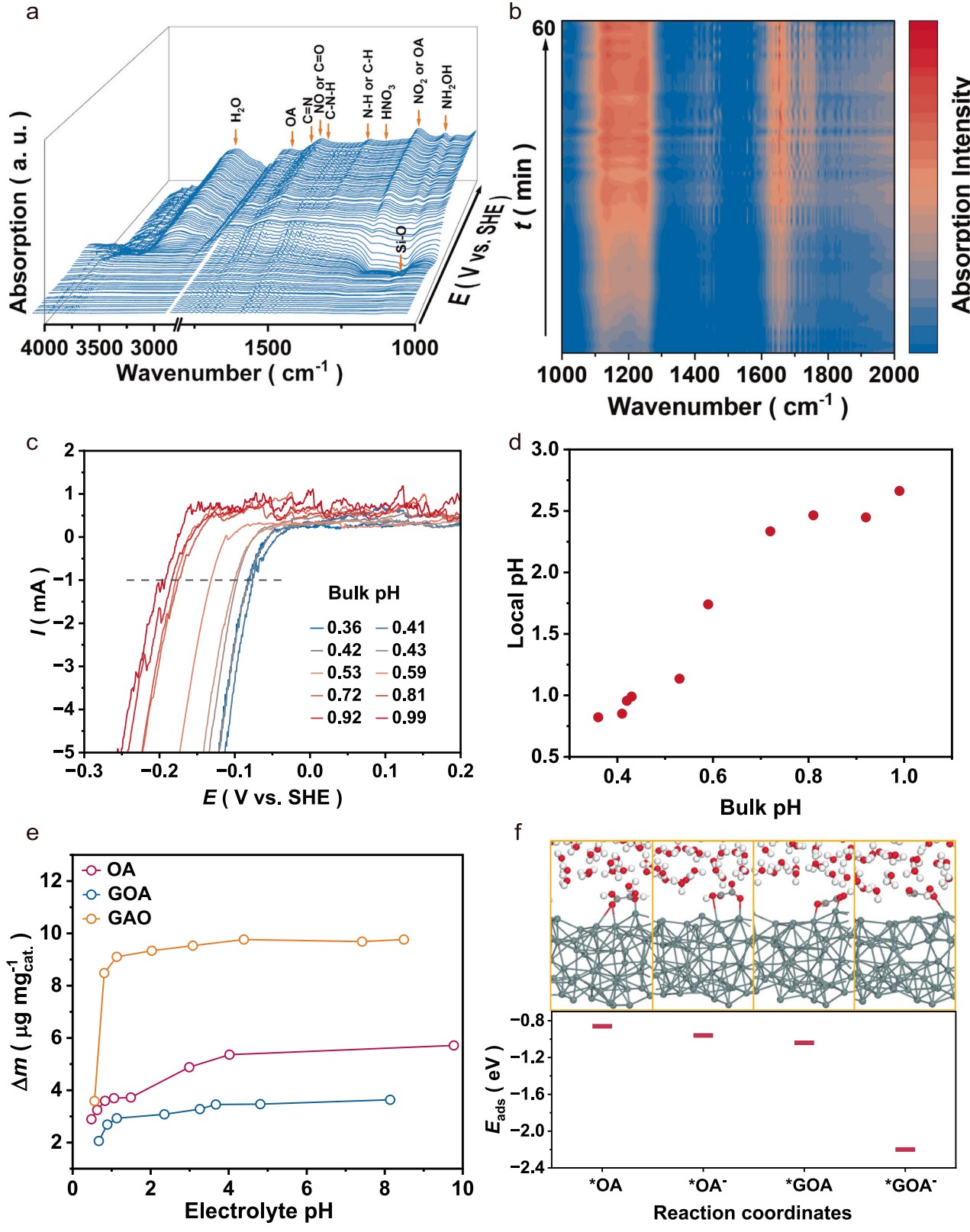

The Faradaic efficiency was calculated according to the following equation:

$$\mathrm{FE_A} = \frac{Q_\mathrm{A}}{Q_\mathrm{total}} = \frac{n_\mathrm{A} V c_\mathrm{A} F}{Q_\mathrm{total}}$$

$$E_\mathrm{RHE} = E_\mathrm{SHE} + 0.0591 \times pH$$

where $E_\mathrm{Ag/AgCl}$ represents the experimental applied potential, and $\phi_\mathrm{reference}$ is 0.198.

**Fig. 5 | Investigation of reactants and microenvironment evolution.**
**a** Electrochemical in situ ATR − FTIR spectra of the Sn catalyst during the electrosynthesis of Gly. The electrochemical test potential ranged from 0 V to −1.6 V vs. SHE. The electrolyte was 5 mL of 0.5 M HNO$_3$ + 0.5 M OA. **b** Time-resolved electrochemical in situ ATR − FTIR spectra for synthetic Gly at −1.4 V vs. SHE, and the interval of the spectra is 1 min. **c** LSV curves of the Pt-ring in a 0.5 M HNO$_3$ + 0.5 M OA mixed solution with different volumes of 5 M KOH added. The disk electrode consists of Sn nanoparticles loaded on a glassy carbon electrode, and the disk potential is −1.4 V vs. SHE. **d** Local pH on the catalyst surface in different bulk pH electrolyte. **e** In situ EQCM test for mass changes (Δ$m$) at −1.4 V vs. SHE in 0.5 M OA, GOA and GAO, respectively. H$_2$SO$_4$ (0.25 M) was added to the initial solution such that the H$^+$ concentration was the same as that in the reaction environment. During the test, 5 M KOH was used to adjust the pH. **f** Adsorption energy of optimized geometries of molecules on amorphous Sn surface. The inserted images show the corresponding adsorption geometries on amorphous Sn. Source data for Fig. 5 are provided as a Source Data file.

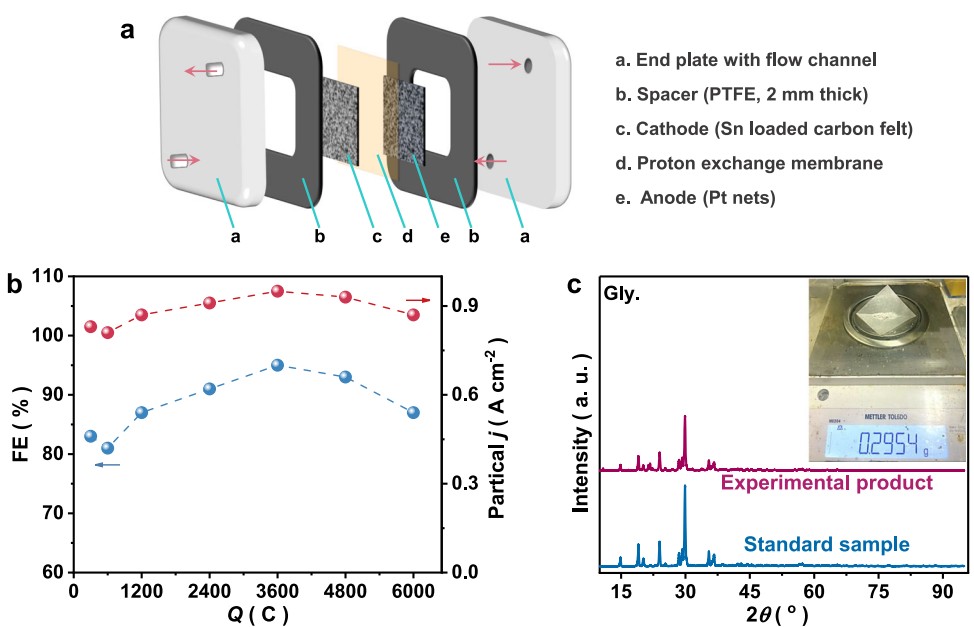

a. End plate with flow channel

b. Spacer (PTFE, 2 mm thick)

c. Cathode (Sn loaded carbon felt)

d. Proton exchange membrane

e. Anode (Pt nets)

**Fig. 6 | Scale-up experiments for the electrosynthesis of Gly. a** Schematic diagram of the internal structure of the flow cell reactor. **b** Coulomb-dependent monitoring results of FE and partial current density of Gly. **c** XRD pattern and photo of the product after one reaction. Source data for Fig. 6 are provided as a Source Data file.

where $Q_{total}$ represents the applied overall coulomb quantity (C). $Q_A$ represents the coulomb required to produce A (A can represent any product, such as glycine, glyoxylic acid, or hydroxylamine et al.). $n$ is the electron transfer number for 1 mol of product A. $V$ is the volume of the catholyte in the cathode chamber, which is 8 mL. $C_A$ is the concentration of the product. $F$ is the Faraday constant (96,485 C mol$^{-1}$).

For the flow-cell measurements (as shown in Fig. 6a), we used chronopotentiometry (10 A) to demonstrate the industrial application potential of the proposed device. The cathode and anode electrolytes are OA + HNO$_3$ and 0.25 M H$_2$SO$_4$, respectively. The flow rate of the electrolyte was 3 to 30 mL min$^{-1}$ (corresponding to a peristaltic pump speed of 10 to 100 rpm).

## In situ XAS experiments
The in situ XAS experiments were performed at P64 beamline at PETRA III synchrotron (Hamburg, Germany)[29]. Sn K-edge data were collected in fluorescence mode using a PIPS detector. A Si(311) double-crystal monochromator was used for energy selection. Intensity of incident X-rays was measured using an ionization chamber filled with 40% Kr in N$_2$ mixture. For operando measurements, we used an in-house-build single-compartment cell, with a leak-free Ag/AgCl reference electrode and glassy carbon counter electrode. Sample was deposited on the GDE, which acted both as a working electrode and the window for incident and fluorescent X-ray photons. During the measurements, Ar was continuously bubbled through the electrolyte, and the continuous electrolyte flow was ensured by a peristaltic pump. Applied potential was controlled by a BioLogic potentiostat. Alignment, normalization and linear combination analysis of XANES data were carried out using

sets of Wolfram Mathematica scripts. EXAFS data were extracted using LARCH code[30].

## Electrochemical in situ Raman tests
The in situ Raman measurements were carried out with the aforementioned Raman microscope and electrochemical workstation. The cell was made up of Teflon with a quartz window between the sample and the objective. The working electrode was immersed in the electrolyte through the wall of the cell, and the electrode plane was kept perpendicular to the laser. A platinum wire and Ag/AgCl served as the counter and reference electrodes, respectively. LSV curves were obtained from 0 to −1.4 V vs. SHE with a scan rate of 2 mV s$^{-1}$. Intermittent electrochemical in situ Raman measurements were carried out in a 0.5 M HNO$_3$ + OA mixed solution at −1.4 V vs. SHE.

## Electrochemical in situ ATR − FTIR tests
Electrochemical in situ ATR − FTIR measurements were performed on a Pike Veemax III ATR with a single-bounce silicon crystal covered with an Au membrane in internal reflection mode. A single−bounce silicon crystal covered with an Au membrane was prepared through the following procedure. (1) NaOH (0.12 g), NaAuCl$_4$·2H$_2$O (0.23 g), NH$_4$Cl (0.13 g), Na$_2$SO$_3$ (0.95 g), and Na$_2$S$_2$O$_3$·5H$_2$O (0.62 g) were dissolved in H$_2$O (100 mL) (denoted as Solution A). (2) Monocrystalline silicon was immersed in aqua regia ($V_{concentrated\ HCl}$:$V_{HNO3}$ = 1:1) for 20 min and then polished with Al powder for 10 min. After washing three times with water and acetone, clean monocrystalline silicon was obtained. (3) The above monocrystalline silicon was immersed in a mixture of H$_2$SO$_4$ and H$_2$O$_2$ ($V_{concentrated\ H2SO4}$:$V_{H2O2}$ = 1:1) for 20 min. (4) After washing

three times with water, the above monocrystalline silicon was immersed in 40% $NH_4F$ aqueous solution and washed three times with water. (5) Monocrystalline silicon was immersed in a mixture of 15 mL of solution A and 3.4 mL of 2% $NH_4F$ aqueous solution. (6) After 5 min, the Au−coated monocrystal silicon was obtained.

## Electrochemical online DEMS test

The electrolyte flowed into a custom-made electrochemical cell through a peristaltic pump. Glassy carbon electrodes coated with the Sn catalyst, Pt wire, and Ag/AgCl electrodes were used as the working electrode, counter electrode, and reference electrode, respectively. Then, an applied potential (−1.4 V vs. SHE) was applied alternately. After the electrochemical test was complete and the mass signal returned to baseline, the next cycle started using the same conditions to avoid accidental error. After several cycles, the experiment ended.

## In situ QECM test

Sn-loaded gold-coated quartz crystals and Ag/AgCl and Pt wires were selected as the working electrode, reference electrode and counter electrode, respectively. The S loading should be less than 0.15 mg. Quartz crystal connection frequency monitoring system. Then, the electrolyte was added to cover the quartz crystal. During electrochemical testing, the crystal vibration frequency can be monitored in real time. The change in mass ($\Delta m$) of the working electrode during electrotesting can be calculated through the Sauerbrey equation:

$$\Delta m = -\frac{\Delta f \times A \times \sqrt{\mu_q \times \rho_q}}{2f_q^2}$$

where $\Delta m$ represents the change mass. $\Delta f$ represents the resonant frequency change. A represents the area of the gold surface (0.198 cm$^2$). $\mu_q$ represents the AT-cut quartz constant ($2.947 \times 10^{11}$ g cm$^{-1}$ s$^{-2}$). $\rho_q$ represents the quartz crystal density (2.65 g cm$^3$). $f_q$ represents the reference frequency (9.00 MHz).

## Determination and quantitation of products via $^1$H NMR

After the electrochemical measurements, 0.5 mL of electrolyte, 50 μL of DIW, and 50 μL of DMSO−d$_6$ were transferred to an NMR tube. The concentration−peak area curve from the NMR spectrum was calibrated via a series of standard sample solutions (5, 10, 15, 20, and 25 mM). For ammonium, we used ammonium chloride as a standard sample, and the ammonium chloride crystals were dried at 105 ∼ 110 °C for 2 h in advance. For the quantification of hydroxylamine, since hydroxylamine does not have the characteristic peak for the NMR spectrum, its quantification requires the preparation of an additional sample to be tested. Specifically, 0.5 mL of electrolyte, 50 μL of GOA, and 50 μL of DMSO−d$_6$ were transferred to another NMR tube. GOA was used to capture hydroxylamine and generate GAO. The concentration of hydroxylamine was subsequently quantified on the basis of a standard curve of GAO. The concentration of hydroxylamine is the same as that of GAO.

## Theoretical calculations

In this work, all theoretical calculations were performed based on spin-polarized density functional theory (DFT) by Vienna Ab initio Simulation Package (VASP)[31,32]. The ion-electron interactions were treated by the projector augmented wave (PAW) method[33]. Generalized gradient approximation (GGA) of the Perdew-Burke Enzerhor (PBE) functional was used to describe the exchange-correlation function[34]. The plane-wave cutoff energy was set to 450 eV and DFT-D3 scheme was employed to consider the long-range van der Waals correction[35]. K-space was sampled by $1 \times 1 \times 1$ Monkhorst-Pack k-point mesh based on the size of the surface supercell. The convergence criteria of energy and force for all structural optimization and energy calculation were set to $10^{-5}$ eV and 0.03 eV Å$^{-1}$, respectively. The

Bader population was used for charge analysis by the code developed by Henkelman et al.[36].

To investigate the energetics of molecule adsorption on Sn catalyst, we performed simulations using low-indexed Sn (101), including crystallized and amorphous ones, which are labeled as Sn(101) and a Sn(101) with six Sn-layers. Given that both OA and GA may dissociate, the calculation of adsorption energy for charged species is not accurate as: (i) OG or GA ions may present different charge states than exactly −1 due to charge redistribution and interfacial bonding; (ii) self-interaction between charged neighboring images can generate errors. Based on this consideration, both molecular and dissociative adsorptions of OA and GA have been investigated on Sn(101) and aSn(101), using the dissociation energy (Ediss = E(dissociative) − E(molecular)) as an indicator, as discussed below. OA and GA have been fully optimized before being introduced to the surface, and the initial configurations have been selected from initial tests with several potential geometries.

## Data availability

The source data underlying Figs. 1–6 and Supplementary Figs. are provided as Source Data. Source data are provided with this paper.

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

## Acknowledgements

K.P. Loh acknowledges support from Singapore's Ministry of Education Tier 1 Grant A80035710000. Kai Leng acknowledges GRF Project PolyU15303024, funded by the Research Grants Council of the Hong Kong Special Administrative Region, China. We acknowledge the National Natural Science Foundation of China (No. 22271213) for financial support. We acknowledge DESY (Hamburg, Germany), a member of the Helmholtz Association HGF, for the provision of experimental facilities. Parts of this research were carried out at PETRA III, and we would like to thank Dr. Aleksandr Kalinko for assistance in using the P64 beamline. Beamtime was allocated for proposal I-20230966. H.L. acknowledges the funding support from the Natural Science Foundation of China (grant no. 22402088) and the Alexander von Humboldt Foundation for a postdoctoral fellowship.

## Author contributions

S. Han and K.P. Loh conceived the idea and supervised the project. S. Han and H. Liu designed the experiments. S. Han synthesized the materials and carried out electrochemical measurements. S. Han and H. Liu designed and performed in situ characterization and control experiments. J. Timoshenko and J. Jang performed the in situ XAS experiments and XAS data analysis. M. Su tested the STEM. C. Sun completed the DFT calculation. S. Han, Y. Yu, and B. Zhang complated in situ XRD and DEMS tests and analysis. Y. Huang, C. Guo, A. Bergmann, B. Roldan Cuenya, Y. Yu, B. Zhang, K. Leng and K.P. Loh revised the manuscript and provided valuable suggestions. S. Han, H. Liu and K.P. Loh wrote the paper with comments from all authors.

## Competing interests

The authors declare no competing interests.
