## [Transparent Peer Review File · Nature Communications]

Sn catalyst reconstruction and microenvironment modulation for efficient amino acid electrosynthesis via C–N coupling

Corresponding Author: Professor Kian Ping Loh

Version 0:

Reviewer comments:

Reviewer #1

(Remarks to the Author)

This manuscript reports an electrosynthetic strategy for amino acid production with good Faradaic efficiency and current density. While the experimental effort and operando characterization are commendable, several key mechanistic claims rely primarily on correlative evidence, and important assumptions regarding active sites, reaction pathways, and mechanism analysis are not acceptable.

1. The proposed transition from a solution-phase chain reaction to an interfacial hydrogenation pathway is mainly inferred from time-dependent concentration trends and adsorption behavior. These observations do not uniquely exclude alternative explanations such as kinetic regime changes or altered adsorption–desorption equilibria. Additional clarification or discussion is needed to justify a true mechanism switch.
2. The manuscript identifies amorphous Sn₀ as the sole active species, yet operando data indicate persistent coexistence of Sn₀, Sn₂₊, and Sn₄₊ under reaction conditions. It remains unclear whether activity arises from amorphous Sn alone or from dynamic Sn/SnO_x interfaces. This conclusion should be better justified or more cautiously framed.
3. The demonstrated spontaneous redox reaction between Sn and HNO₃ raises concerns regarding non-electrochemical contributions to product formation. The authors should clarify how such chemical reactions are excluded from Faradaic efficiency calculations, particularly during the catalyst reconstruction stage.
4. Local pH values are estimated using RRDE measurements with a Pt ring electrode. Given the high current densities and expected strong gradients, the representativeness and spatial resolution of this approach should be discussed. Clarification of potential uncertainties in the local pH estimation is necessary.
5. The mechanism relies on weak acid reactants and intermediates existing predominantly in anionic form due to local pH elevation. However, the reported local pH remains strongly acidic. Additional quantitative justification or discussion is needed to support the dominance of anionic species at the catalyst interface.
6. Hydroxylamine concentration is indirectly determined via trapping with glyoxylic acid. Given the instability and reactivity of NH₂OH in acidic media, the accuracy and potential limitations of this quantification method should be addressed.
7. The DFT calculations acknowledge limitations in treating charged adsorbates and rely on dissociation energy trends. These results should be clearly positioned as qualitative support rather than decisive mechanistic proof.
8. The manuscript emphasizes the unique role of Sn under nitric acid conditions, but it remains unclear whether the proposed mechanism is Sn-specific or extendable to other dynamically reconstructing metals. A brief discussion of this point would enhance the generality of the study.
9. Although α -keto acids are successfully converted, simple carboxylic acids are inactive, indicating reliance on pre-existing carbonyl functionality. The authors should clarify how this limitation affects the claimed generality of the C–N coupling strategy.
10. Continuous redox cycling and spontaneous Sn oxidation in nitric acid may affect long-term catalyst stability. The authors should discuss expected catalyst lifetime and potential implications for continuous operation.
11. The techno-economic analysis appears promising but does not fully account for costs related to nitric acid handling, NO_x management, and catalyst degradation. Clarification of key assumptions and uncertainties is recommended.

Reviewer #2

(Remarks to the Author)

The authors have addressed most comments sufficiently. I can at this point support the publication of this work.

Reviewer #3

(Remarks to the Author)

All concerns have been well addressed, and I recommend its publication at its current form.

Version 1:

Reviewer comments:

Reviewer #1

(Remarks to the Author)

I have no further comments on the revised paper.

A point-by-point response to the reviewer's comments

To Reviewer 1:

Reviewer Letter: This manuscript reports an electrocatalytic strategy for amino acid production with good Faradaic efficiency and current density. While the experimental effort and operando characterization are commendable, several key mechanistic claims rely primarily on correlative evidence, and important assumptions regarding active sites, reaction pathways, and mechanism analysis are not acceptable.

Answer: Thank you for acknowledging the value of our work. We have responded to your suggestions point by point. We hope these explanations address your concerns. Based on your recommendations, we have revised both the manuscript and the supporting information. To save your time, all changes have been highlighted in yellow.

Comment 1: The proposed transition from a solution-phase chain reaction to an interfacial hydrogenation pathway is mainly inferred from time-dependent concentration trends and adsorption behavior. These observations do not uniquely exclude alternative explanations such as kinetic regime changes or altered adsorption–desorption equilibria. Additional clarification or discussion is needed to justify a true mechanism switch.

Answer 1: Thank you for your insightful feedback. We agree that reaction kinetics and adsorption–desorption equilibria evolve during the reaction, but we do not believe this contradicts our proposed mechanism shift from a solution-phase chain process to interfacial hydrogenation. The two mechanisms share a common reaction pathway but yield significantly different outcomes—particularly in the faradaic efficiency (FE) of glycine. A purely chain mechanism would maintain a steady intermediate concentration in solution, regardless of kinetic variations, and could not account for the observed FE increase. This is why prior studies (e.g., *Nat. Catal.*, 2023, 6, 906–915) used a two-step tandem approach with isolated intermediate synthesis.

Rapid intermediate readsorption—with negligible diffusion into the electrolyte—is unlikely, as strongly adsorbed species tend not to desorb easily. Even if it occurred, the resulting behavior would mirror interfacial hydrogenation, making it an apparent interfacial mechanism. For clarity, we have refined both mechanism definitions in the revised Supplementary Note 5.

Our subsequent in-situ studies confirm that the transition is driven by changes in intermediate adsorption–desorption equilibrium, induced by catalyst amorphization and local pH increases. These alter adsorption energies, favoring complete hydrogenation at the surface.

Current literature often conflates these pathways, with some studies implying chain processes while analyzing interfacial mechanisms. Our work aims to clarify this distinction and offer guidance for the field.

*“Notably, the above explanation is an ideal situation. In actual catalytic processes, the two mechanisms may coexist, or different mechanisms may be exhibited for different reactants. For example, in our system, the key intermediate NH_2OH is desorbable in both mechanisms. In this work, the judgment of these two mechanisms is based on the evolutionary process of C-species, because the conversion of C-species is the rate-limiting step in the synthesis of Gly (evidence will be provided below). Considering the limitations of experimental detection sensitivity, we distinguish between the two mechanisms here based on the NMR results of the intermediate concentration. When the desorption amount of GOA and GAO intermediates is below the NMR detection limit, we consider it to have no apparent desorption behavior. The C-N coupling process likely involves the desorbed NH_2OH acting as a nucleophile to couple with the adsorbed *GOA at the Sn surface, transforming it into *GAO. There may be other complex processes involved, but since these cannot be determined experimentally, we will not speculate further.”*

Comment 2: The manuscript identifies amorphous Sn⁰ as the sole active species, yet operando data indicate persistent coexistence of Sn⁰, Sn²⁺, and Sn⁴⁺ under reaction conditions. It remains unclear whether activity arises from amorphous Sn alone or from dynamic Sn/SnO_x interfaces. This conclusion should be better justified or more cautiously framed.

Answer 2: Thank you for raising this important point. We acknowledge the coexistence of both a spontaneous chemical redox reaction and an electrochemical reduction pathway in our system. We chose Sn⁰ as the active species based on subsequent Bode plots and electrochemical control experiments. Using SnO or SnO₂ as pre-catalysts, their yield rate of Gly is significantly lower than that of Sn⁰. Considering that they all should reconstruct in situ to form the Sn/SnO_x interface during electrochemical processes, although the main components might be SnO_x, we hypothesized that Sn⁰ was more likely the true active site. Based on your suggestion, we revised the relevant wording, using more cautious terminology to express our inferences in revised manuscript and revised Supplementary Information (Supplementary Note 9).

The key distinction lies in the role of tin: it serves as an electron mediator, undergoing a cyclic valence change ($\text{Sn}^0 \rightarrow \text{Sn}^{\text{II}} \rightarrow \text{Sn}^0$) between metallic tin and SnO. Although a small amount of SnO is consumed chemically, its contribution is negligible relative to the total charge passed during glycine electrosynthesis. Electrons are ultimately sourced from the electrode, not from the chemical reaction.

“Amorphization alters both valence states and structural configurations. For different valence, the Bode plot of standard sample showed the phase angle and frequency of Sn foil and SnO were similar to our catalyst. Low phase angle and high frequency suggest a faster electron exchange rate. The LSV curve revealed that j of Sn foil was close to the experimental results (Supplementary Figure 4) and significantly larger than those of SnO and SnO₂ (Supplementary Figure 27), ruling out the possibility of SnOx as an active site. Based on the results of in situ Raman and in situ XAS, all three catalysts can undergo reconstruction during the electrochemical process, forming Sn/SnOx interfaces, albeit with different bulk compositions. Therefore, from a qualitative analysis perspective, the in situ constructed Sn/SnOx sites may not be the most effective sites. Thus, Sn⁰ may be the main active site, and the amorphous structure may be more conducive to improving the electron transfer rate (Figure 4g).”

Comment 3: The demonstrated spontaneous redox reaction between Sn and HNO₃ raises concerns regarding non-electrochemical contributions to product formation. The authors should clarify how such chemical reactions are excluded from Faradaic efficiency calculations, particularly during the catalyst reconstruction stage.

Answer 3: We fully understand the reviewer's concerns. This process involves two pathway: a non-electrochemical redox reaction and a catalytic electroreduction process. For the Sn valence, in the electrochemical process, Sn reacts spontaneously with the reactant HNO₃ to form SnO. Simultaneously, the generated SnO undergoes electroreduction to a metallic state, completing the valence state cycle of the catalyst (Sn⁰—Sn^{II}—Sn⁰). Therefore, the catalyst Sn acts as an electron mediator, and its valence remains essentially unchanged before and after a reaction cycle (the small amount of SnOx consumed is negligible compared to the coulombic amount required for the electrosynthesis of glycine). Electrons are ultimately transferred to HNO₃. Therefore, this mechanism difference compared to other reports does not affect the FE calculation.

Comment 4: Local pH values are estimated using RRDE measurements with a Pt ring electrode. Given the high current densities and expected strong gradients, the representativeness and spatial resolution of this approach should be discussed. Clarification of potential uncertainties in the local pH estimation is necessary.

Answer 4: Thanks for the reviewers' suggestions. We agree with your assessment. Due to there is a gap (~0.2 mm) between the Pt ring and the glassy carbon electrode (working electrode), there is

some error in measuring the pH of the catalyst surface using RRDE. This is due to the limitations of the RRDE device. To ensure scientific rigor, we have included appropriate error annotations in the revised Supplementary Information (Supplementary Note 11). In the field of electrochemistry, the use of RRDE to estimate the local pH of a reaction is accepted (e.g., CO₂RR, HER et al.), but it has not yet been explored in the C-N coupling field. However, they work on similar principles. Here, the Nernst equation for Pt at different H⁺ concentrations is used to calculate the local pH. Since the Pt ring is very close to the working electrode, the local pH of the Pt ring can be approximated as the pH value of the working electrode when the electrode is rotated rapidly (1600 rpm).

“It should be noted that this method may have certain limitations, due to the 0.2 mm gap between the Pt ring and the glassy carbon electrode (carried catalyst) of RRDE. Although the tests were conducted at a high rotation speed of 1600 rpm, diffusion limitations may cause the local pH of the Pt ring to be lower than that of the catalyst interface.”

Comment 5: The mechanism relies on weak acid reactants and intermediates existing predominantly in anionic form due to local pH elevation. However, the reported local pH remains strongly acidic. Additional quantitative justification or discussion is needed to support the dominance of anionic species at the catalyst interface.

Answer 5: We fully understand the reviewers' concerns. The form in which weak electrolytes exist (molecules or ions) depends on the relationship between their own ionization constant (such as pK_a or pK_b) and the solution pH. The specific amount of ionization in the solution can be calculated using the Henderson-Hasselbalch equation. Based on the test results, the pH of the mixed solution before the reaction is 0.45, and the pH of the local microenvironment during the reaction process is ~2.5. The pH values of 0.5M weak acid solutions of OA, GOA, and GAO are 0.81, 0.99, and 1.23, respectively. Taking OA as an example:

$$\text{H}^+ \approx 0.15 \text{ M}$$

$$\text{A}^- \approx 0.15 \text{ M}$$

$$\text{HA} = 0.5 - 0.15 = 0.35 \text{ M}$$

$$K_a = \frac{[\text{H}^+] \times [\text{A}^-]}{[\text{HA}]}$$

So:

$$\text{pK}_a \approx 1.19$$

Before the reaction, the pH of the mixed solution was 0.45, as the main contribution came from the ionization of HNO₃ (strong electrolyte). According to the Henderson-Hasselbalch equation:

$$pH = pKa + \log \frac{[A^-]}{[HA]}$$

$$HA \approx 0.42 \text{ M}$$

$$A^- = 0.5 - 0.42 = 0.08 \text{ M}$$

HA represents the molecular form of OA. A⁻ represents the ionic form of OA. H⁺ is the hydrogen ion. We can see that most weak acids exist in molecular form.

Similarly, during the catalytic process, the local pH is 2.5. According to the Henderson-Hasselbalch equation:

$$pH = pKa + \log \frac{[A^-]}{[HA]}$$

$$HA = 0.023 \text{ M}$$

$$A^- = 0.5 - 0.023 = 0.477 \text{ M}$$

Notably, since the in-situ catalytic process is relatively short, the concentration of OA is calculated at 0.5 M. At this point, over 95% of the OA in the microenvironment exists in ionic form.

The existence forms of GOA and GAO are calculated using a similar method. The results showed that the ion concentrations of GOA and GAO during the electrocatalytic process were 0.44 and 0.36 M, respectively. Therefore, the actual form in the microenvironment may be ions rather than molecule.

To address the concerns of reviewers and readers, we have added the above calculation process to Supplementary Note 13.

Comment 6: Hydroxylamine concentration is indirectly determined via trapping with glyoxylic acid. Given the instability and reactivity of NH₂OH in acidic media, the accuracy and potential limitations of this quantification method should be addressed.

Answer 6: Thanks for the reviewer's suggestions. According to recent reports, there are two methods for detecting hydroxylamine: UV-vis spectrophotometry and NMR spectroscopy. Both methods require the preparation of a standard curve, and then calculate the hydroxylamine concentration in the sample based on it. Due to the various substances in the C–N coupling system, UV-vis spectrophotometry is not suitable for hydroxylamine detection. For NMR detection methods,

since hydroxylamine does not have characteristic ^1H NMR signals, researchers typically add an aldehyde or ketone (such as formaldehyde or acetone) to capture hydroxylamine and convert it into an oxime, which has distinct ^1H NMR signals. Notably, oximes are significantly more stable than hydroxylamine, so there is no concern about their decomposition during the testing process. Furthermore, the quantitative analysis of the samples is based on the NMR standard curve, and the operating procedures are same. Therefore, even if minor conversion occurs, it does not affect the accuracy of the quantitative results. In our system, GOA is involved, so it can be directly used as a trapping agent to capture excess hydroxylamine. Therefore, this method is reliable. We added a note in the Supporting Information (Supplementary Note 3) to explain this point.

Comment 7: The DFT calculations acknowledge limitations in treating charged adsorbates and rely on dissociation energy trends. These results should be clearly positioned as qualitative support rather than decisive mechanistic proof.

Answer 7: We appreciate the reviewer's careful and insightful comment regarding the limitations of DFT calculations in treating charged adsorbates. We fully agree that, due to well-known challenges associated with charged species under periodic boundary conditions, the present DFT results should not be interpreted as decisive mechanistic proof.

In the revised manuscript, we have therefore explicitly clarified that the DFT calculations are intended to offer qualitative support by capturing relative trends in dissociation preference and adsorption behavior, rather than serving as a quantitative or definitive description of the reaction mechanism.

Importantly, the proposed mechanistic picture is primarily established based on extensive experimental evidence, including time-resolved NMR, in situ XRD, Raman, XAS, etc. Under such scheme, DFT results are employed to rationalize and corroborate these experimentally observed trends.

To avoid any possible overinterpretation, we have revised the corresponding discussion to explicitly emphasize the qualitative nature of the theoretical analysis and its complementary role alongside the experimental findings.

“Due to the intrinsic limitations of modeling charged adsorbates within periodic DFT, the calculated dissociation energies are used to indicate qualitative trends rather than to determine absolute reaction energetics.”

Comment 8: The manuscript emphasizes the unique role of Sn under nitric acid conditions, but it remains unclear whether the proposed mechanism is Sn-specific or extendable to other dynamically reconstructing metals. A brief discussion of this point would enhance the generality of the study.

Answer 8: Thanks for the reviewer's suggestions. We agree with your views. We believe that Pb should show similar properties as it is in the same group of the periodic table as Sn. From a thermodynamic perspective, elements with standard reduction potentials similar or negative to that of Sn are likely to undergo similar reaction mechanisms. Based on your suggestion, we have added a brief discussion in revised Supplementary Information to encourage researchers to consider similar mechanisms when investigating other elemental catalysts.

“From a thermodynamic perspective, elements with standard electrode potentials similar or negative to that of Sn may undergo the aforementioned catalytic valence cycle mechanism in an acidic electrosynthesis glycine system, highlighting the importance of monitoring catalyst dynamic reconstruction to reveal the true active sites.”

Comment 9: Although α -keto acids are successfully converted, simple carboxylic acids are inactive, indicating reliance on pre-existing carbonyl functionality. The authors should clarify how this limitation affects the claimed generality of the C–N coupling strategy.

Answer 9: Thank you for the suggestion. We have revised the manuscript to emphasize that the scope is limited. While electrosynthesis of amino acids is a growing field, few studies explore method generalizability. Some reports address this using α -keto acids—a common reactant class due to their suitability for amino acid synthesis. Our system also demonstrates general applicability with α -keto acid reactants.

We further tested non- α -keto acid carbon sources. Although ketones and aldehydes can theoretically be transformed into amines, they instead formed oximes—a relevant outcome for researchers in C–N coupling. In contrast, ordinary carboxylic acids did not react, as our Sn sites cannot electro-reduce -COOH groups lacking an α -carboxyl. This reaction would require additional catalytic sites capable of reducing -COOH to -C=O. By including both successful and unsuccessful outcomes, we provide a comprehensive and accurate account of our work, which may benefit researchers in electrosynthesis of amino acids, oximes, and amines.

Comment 10: Continuous redox cycling and spontaneous Sn oxidation in nitric acid may affect long-term catalyst stability. The authors should discuss expected catalyst lifetime and potential implications for continuous operation.

Answer 10: Thanks for the reviewer's suggestion. We agree with the reviewer's hypothesis. This reaction mechanism may affect the long-term stability of the catalyst. As we presented in the stability test in the manuscript, it is 70 hours. Longer time test results showed that the performance began to gradually decline. However, this stability is significantly higher than the benchmark performance in recent reports. Based on your suggestion, we have added a discussion of the stability in the revised manuscript.

“Notably, this catalyst reconstruction mechanism may affect electrocatalyst stability. The inability to complete a valence cycle or the loss of Sn element after a long-term testing may lead to its stability decrease.”

Comment 11: The techno-economic analysis appears promising but does not fully account for costs related to nitric acid handling, NO_x management, and catalyst degradation. Clarification of key assumptions and uncertainties is recommended.

Answer 11: Thank you for this important feedback. Our techno-economic analysis (TEA) follows established methodologies (e.g., Nat. Synth., 2025, DOI: 10.1038/s44160-025-00892-7) and is intended primarily to enable comparative assessment within the research community and highlight the field's potential, not to precisely predict real-world costs.

We fully acknowledge that practical considerations such as nitric acid handling, NO_x management, and catalyst durability introduce significant uncertainties. However, due to the absence of experimental data or industrial references for these aspects, their impacts could not be quantitatively incorporated. We have added a clarification in the Supplementary Information to explicitly note these assumptions and limitations.

We appreciate your emphasis on these critical factors and agree that they warrant attention in future applied studies.

“Notably, this is just a simplified calculation model for easy comparison with other reports. Actual industrial production may differ significantly. The costs of the handling of residual reactants, NO_x management, and catalyst degradation et. al, which may be involved in practical applications, are not considered in this paper due to the lack of actual industrial application parameters.”

The Hong Kong Polytechnic University
National University of Singapore

E-mail: chmlhkp@nus.edu.sg

We thank the four reviewers for their kind and professional suggestions. We are sure that the quality of this work will be greatly improved according to these helpful comments and suggestions from the four reviewers.